# Learned Sequence Representations over Raw Credit Events for Credit-Abuse Scoring

Tianming Zhou[1]   Jiarui Xu[1]   Nitesh Kumar[1]   Alexander Statnikov[1]

## Abstract

Credit abuse is a rare and costly outcome at consumer-credit platforms; industry evidence places its concentration in moderately-aged credit files. Tuned gradient-boosted decision trees (GBDTs) over hand-engineered features are the prevailing choice, and recent tabular foundation models are advertised to match them. We ask whether a learned representation over raw credit-event sequences, used in place of engineered aggregates of those events, is a better fit. To produce this representation, we pretrain a causal Transformer on credit-event sequences with self-supervised objectives and adapt the backbone to the downstream label via supervised finetuning. On an out-of-time evaluation, under both an XGBoost classifier and an in-context tabular foundation model, the sequence-based feature set beats the engineered-aggregate alternative on AUPR (XGBoost: 0.068 → 0.147; TabPFN: 0.097 → 0.141). Among single-factor upgrades from the (XGBoost, engineered) baseline, at production training scale the representation-only upgrade (XGBoost on the sequence-based vector) outperforms the classifier-only upgrade (TabPFN on the engineered alternative) on both AUPR (0.147 vs 0.097) and recall at a 0.5% decline threshold (0.230 vs 0.184).

## 1. Introduction

Gradient-boosted decision trees (GBDTs), especially XGBoost (Chen & Guestrin, 2016), have long been the default scoring model in consumer-credit underwriting. They train cheaply, are well-calibrated out of the box, and consume the engineered feature sets that credit-risk teams maintain. The past two years have brought a wave of *tabular foundation models* — TabPFN (Hollmann et al., 2023; 2025), TabICL

(Qu et al., 2025), TabDPT (Ma et al., 2025), Mitra (Zhang et al., 2025) — and a prior-literature claim, consolidated in the TabArena benchmark (Erickson et al., 2025), that they now approach or match GBDTs on standard tabular tasks. We test whether that claim carries over to a production credit-abuse setting and — more centrally — whether a learned representation over *raw* credit-event sequences improves performance over engineered aggregates of the same events, under either classifier (GBDT or tabular FM).

**Credit abuse** is a rare and costly early-life default label that, in industry reporting, concentrates in the early years of an applicant's credit history — after enough credit-file activity has accumulated to support fraud-ring usage but before the file matures into a stable repayment record (TransUnion, 2025; SentiLink, 2024). The scoring-design question for abuse is not "does the model see enough features" — engineered feature sets already summarize every signal the platform has — but "does the model see the *ordered event history* over which the abuse lifecycle unfolds."

We cast this as two testable hypotheses.

**(H1) Sequence-based representations carry signal beyond engineered aggregates for credit-abuse prediction.** Replacing engineered aggregates with a learned sequence-based representation over the same events improves performance under any classifier.

**(H2) Representation is the load-bearing factor.** From a (GBDT, engineered) baseline, switching the engineered aggregate for a learned sequence-based representation improves performance more than switching the GBDT for an in-context tabular FM. Practically: for a fixed upgrade budget, invest in representation before classifier.

**Contributions.** 1. **An architecture and training recipe** for credit-event sequences: a compact tokenization that encodes 13 heterogeneous credit-event types — each with its own feature schema — through per-event-type MLP encoders into a shared model dimension, plus a Time2Vec temporal embedding for irregular event timing; self-supervised pretraining; and supervised finetuning on the downstream credit-abuse label. 2. **A controlled experimental design** in which the engineered-aggregate baseline and the sequence-based feature set see the same underlying signals and differ

[1]Affirm, Inc.. Correspondence to: Alexander Statnikov <alexander.statnikov@affirm.com>.

*Proceedings of the 2nd ICML Workshop on Foundation Models for Structured Data*, Seoul, South Korea. 2026. Copyright 2026 by the author(s).

only in how the event stream enters — as scalar aggregates or as a learned sequence representation. This is the step that lets the experimental gap between the two feature sets be read as the value of *structure* (order, timing, co-occurrence) in the raw event stream. 3. **Evidence on credit abuse** that the sequence-based representation significantly outperforms engineered aggregates (H1) and that the representation upgrade dominates the classifier upgrade (H2), at $p<0.05$ on out-of-time evaluation.

## 2. Related Work

**Tabular foundation models.** TabPFN (Hollmann et al., 2023; 2025) pretrains a Transformer on synthetic tabular priors and serves predictions via a single in-context forward pass. TabICL (Qu et al., 2025) scales in-context learning to 500k-row tables using a column-then-row attention stack. TabDPT (Ma et al., 2025) pretrains on real tables. Mitra (Zhang et al., 2025) mixes synthetic priors. Semantic and heterogeneous variants include CARTE (Kim et al., 2024), TabSTAR (Arazi et al., 2025), and ConTextTab (Spinaci et al., 2025). The TabArena benchmark (Erickson et al., 2025) consolidates this family and reports that GBDTs remain strong overall, with tabular FMs most competitive at small sample sizes. We evaluate TabPFN as a representative in-context tabular FM in its canonical inference mode.

**Transformers for credit and financial sequences.** PRAGMA (Ostroukhov et al., 2026) pretrains a Transformer on transaction sequences with a self-supervised masked objective, with downstream uses including credit risk. Fin-LangNet (Lei et al., 2026) applies language-modeling-style self-supervision to credit-risk sequences. nuFormer (Braithwaite et al., 2025) integrates learned transaction embeddings with engineered tabular features through end-to-end joint-fusion fine-tuning. Broader sequence-FM precedents such as MOMENT (Goswami et al., 2024) and Chronos (Ansari et al., 2024) establish the same self-supervised-pretrain-then-finetune recipe in the time-series domain. Together these support our framing that sequence-aware self-supervision can produce useful representations of applicant histories. Our tokenization handles a multi-type credit-event vocabulary in a single sequence, and we follow pretraining with supervised finetuning on the downstream credit-abuse label before stacking against an engineered complement, in the wide-and-deep style of (Cheng et al., 2016).

## 3. Method

### 3.1. Problem setup

**Deployment context.** The model is invoked at *scoring time* — the moment an applicant submits a loan application — and produces a continuous risk score (the predicted probability of credit abuse). At each scoring event we observe the

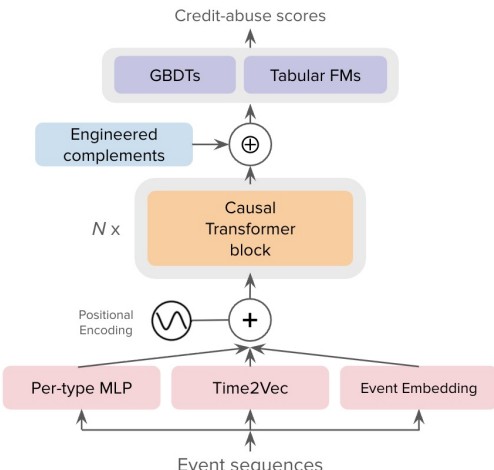

*Figure 1. Backbone + feature pipeline. Per-event-type MLPs, a shared Time2Vec temporal embedding, and a learned event-type embedding are summed and passed through a causal Transformer. The last-position hidden state is concatenated with an engineered complement, producing the sequence-based feature vector scored by a binary classifier.*

applicant's variable-length history of **credit events** up to scoring time, drawn from a 13-type vocabulary covering the applicant's prior interactions with the scoring platform and their broader credit-history records. The target is a binary credit-abuse label for the current scoring event (concrete observation gap in Appendix B).

**Study population.** Mid-tenure applicants with credit-file age between 36 and 60 months, the segment with enriched credit-abuse prevalence (§1).

**Data splits.** Applicants are partitioned into two disjoint cohorts; only one (the *training* cohort) contributes rows to `train`. Out-of-time evaluation uses two splits: `OOT-in-cohort` (training-cohort applicants in the OOT window — tests temporal generalization) and `OOT-out-of-cohort` (held-out-cohort applicants in the OOT window — tests temporal *and* user-cohort generalization). The pooled `OOT-merged` is the primary evaluation; split statistics and positive counts in Appendix B.

### 3.2. Tokenization and backbone

Each position carries an event-type id $e_t \in \{0, \dots, 12\}$ (13 event types) and a feature vector whose first two channels are an absolute timestamp and the time-delta since the previous event. Three additive embeddings are summed at each position: a per-event-type MLP encodes the type-specific channels; a shared Time2Vec (Kazemi et al., 2019) module encodes the timestamp pair; a learned event-type embedding identifies the event type at positions with empty schema.

The encoder is a **6-layer causal Transformer** (Figure 1); full hyperparameters in Appendix A.

*Table 1. Credit-abuse AUPR and recall@0.5% on `OOT-merged` with 95% CIs. Best per row in bold; higher is better.*

| Metric | Classifier | Engineered aggregates | Sequence-based representation |
|---|---|---|---|
| AUPR | XGBoost | 0.068 [0.044, 0.106] | **0.147 [0.105, 0.201]** |
| | TabPFN (in-context) | 0.097 [0.066, 0.142] | **0.141 [0.095, 0.197]** |
| recall@0.5% | XGBoost | 0.124 [0.083, 0.171] | **0.230 [0.180, 0.286]** |
| | TabPFN (in-context) | 0.184 [0.134, 0.235] | **0.230 [0.175, 0.281]** |

*Table 2. One-sided 95 % paired-bootstrap differences on `OOT-merged`. $\Delta = A - B$; lower bound > 0 indicates $p < 0.05$.*

| Cell A | Cell B | $\Delta$ AUPR [LB, UB] | $\Delta$ recall@0.5% [LB, UB] |
|---|---|---|---|
| XGBoost + Sequence | XGBoost + Engineered | +0.079 [+0.046, +0.107] | +0.106 [+0.065, +0.147] |
| TabPFN + Sequence | TabPFN + Engineered | +0.044 [+0.015, +0.070] | +0.046 [+0.009, +0.083] |
| XGBoost + Sequence | TabPFN + Engineered | +0.051 [+0.018, +0.078] | +0.046 [+0.009, +0.088] |

### 3.3. Self-supervised pretraining

**Next-event-type prediction.** A linear head over the 13-event-type vocabulary is attached at every position; the model is trained to predict $e_{t+1}$ from the causal hidden state at position $t$. The loss is masked cross-entropy over all non-padding, non-terminal positions:

$$\mathcal{L}_{\text{next}} = \frac{1}{|\mathcal{P}|} \sum_{t \in \mathcal{P}} \text{CE}\big(e_{t+1},\ \text{softmax}(W_{\text{next}} h_t)\big).$$

**Temporal ranking.** For each position $t$ we form one positive candidate — the immediately-next event at $t + 1$ — and $K$ negative candidates at randomly-sampled later positions $t + k$ ($k \geq 2$) drawn from a truncated geometric distribution. A bilinear ranking head $r_\phi(h_t, x_{t+\Delta})$ maps the causal hidden state $h_t$ and a candidate input embedding $x_{t+\Delta}$ to a scalar score; a margin ranking loss enforces a higher score for the positive ($\Delta = 1$) than for each negative ($\Delta = k \geq 2$):

$$\mathcal{L}_{\text{TR}} = \frac{1}{|\mathcal{N}|} \sum_{(t,k) \in \mathcal{N}} \big[\delta - r_\phi(h_t, x_{t+1}) + r_\phi(h_t, x_{t+k})\big]_+.$$

**Joint loss.** The total pretraining loss is $\mathcal{L}_{\text{pre}} = \mathcal{L}_{\text{next}} + \lambda\,\mathcal{L}_{\text{TR}}$ with $\lambda = 0.3$. Hyperparameters in Appendix A.

### 3.4. Supervised finetuning

The pretrained backbone is further trained end-to-end on the credit-abuse label. The next-event and temporal-ranking heads are replaced with a lightweight binary-classification head applied to the hidden state at the last sequence position; loss is focal binary cross-entropy with negative downsampling. Hyperparameters in Appendix A.

### 3.5. Feature sets

The Table 1 panel compares two feature sets that share a 71-d engineered complement and differ only in how the within-platform event stream enters — as a learned Transformer embedding or as engineered scalar aggregates.

**Shared 71-d engineered complement.** External-bureau summary features and applicant-level context. The complement deliberately excludes engineered aggregates of the within-platform events, so that the within-platform stream is the only axis along which the two feature sets differ.

**Sequence-based representation (199 d).** After finetuning, the classification head is discarded, the backbone is frozen, and the forward pass terminates at the last-position hidden state. The resulting 128-d embedding is concatenated with the 71-d complement.

**Engineered aggregates (199 d).** The 71-d complement is extended with 128 within-platform engineered aggregates from the full internal pool, matching the cardinality of the sequence-based feature set.

## 4. Experiments

### 4.1. Setup

**Metrics.** Primary: **AUPR** on the credit-abuse label, well-suited to the rare-label regime. Secondary: **recall@0.5%** — recall at the 0.5 % decline operating point.

**Statistical protocol.** Marginal CIs (Table 1) are 95 % class-stratified bootstrap. H1 and H2 are directional, so paired comparisons (Table 2) use the **one-sided 95 % paired bootstrap** on the same resampled rows: we report the bracket [LB, UB] as the 5th–95th percentiles of the bootstrap distribution; a lower bound > 0 is reported as p < 0.05.

**Feature sets.** Both 199-d, defined in §3.5; both share the 71-d engineered complement and differ only in how the

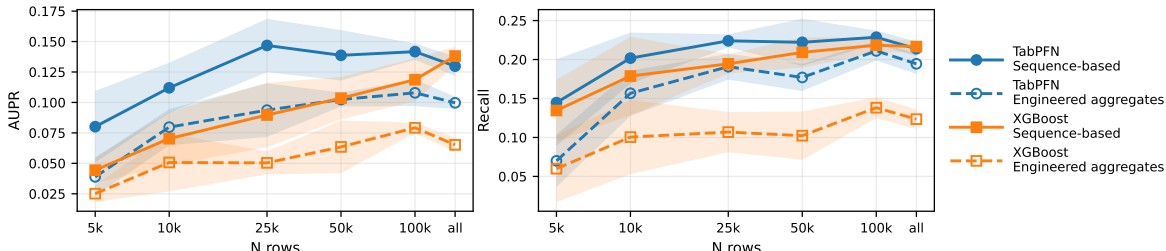

*Figure 2.* Data-efficiency sweep on `OOT-merged`. X-axis: $n$ of downstream training rows (log scale). Y-axis: AUPR (left) and recall@0.5% (right). Marker shape encodes classifier (circle = TabPFN, square = XGBoost); marker fill encodes feature set (filled = Sequence-based, unfilled = Engineered aggregates). Bands: $\pm 1$ SD across 5 stratified seeds.

within-platform event stream enters.

**Classifiers.** Two per feature set. (i) **XGBoost** at the production-pipeline configuration for similar tasks (Appendix A), unchanged across feature sets to isolate representation from tuning effects. (ii) **TabPFN** (Hollmann et al., 2025) in canonical in-context mode (TabPFN v3). Fine-tuned and HP-search variants of TabPFN and Mitra are evaluated as ablations in Appendix C.3.

### 4.2. Main result

Tables 1 and 2 establish two findings.

**F1 (tests H1) — Sequence > Engineered.** Under either classifier, the sequence-based feature set significantly outperforms the engineered alternative on `OOT-merged` for both metrics (Table 2 rows 1–2), with all four comparisons p < 0.05 paired bootstrap. Under the controlled design of §3.5, this gap measures the value of *structure* in the raw event stream (order, timing, co-occurrence) that scalar aggregates discard.

**F2 (tests H2) — Representation upgrade dominates classifier upgrade as a single-factor pick from the baseline.** Starting from the (XGBoost, engineered) configuration, the representation-only upgrade (XGBoost on the sequence-based vector) reaches AUPR 0.147 / recall 0.230 — outperforming the classifier-only upgrade (TabPFN on the engineered alternative, AUPR 0.097 / recall 0.184) on both metrics (Tables 1, 2).

### 4.3. Data efficiency

Figure 2 plots the data-efficiency sweep. We hold the task-adapted backbone frozen and vary the downstream classifier's training-subset size $n \in \{5k, 10k, 25k, 50k, 100k, \text{full}\}$ via stratified sub-sampling, retraining each configuration with 5 seeds at each $n$.

**Sequence-based > Engineered aggregates at every $n$** under both classifiers — H1 is robust to training-pool size, not an artifact of the $n = \text{full}$ operating point.

**TabPFN+Sequence dominates the small-to-mid-$n$ regime.** It is the strongest configuration at every $n$ from 10k to 50k, peaking at $n = 25k$ at approximately $1.6\times$ the same-$n$ XGBoost+Sequence AUPR (0.090). Below $n = 10k$ the rare-event noise floor takes over: at $n = 5k$ each stratified seed sees $\approx 9$ positives, and all four configurations converge to AUPR $\leq 0.08$ with wide CIs.

**F2 is a large-$n$ AUPR phenomenon.** At small $n$ the classifier-only upgrade (TabPFN+Engineered) edges the representation-only upgrade (XGBoost+Sequence) at $n = 10k$ (0.080 vs 0.070) — consistent with the canonical small-$n$ strength of tabular FMs — they reach parity at $n = 25k$ (0.094 vs 0.090), and XGBoost+Sequence overtakes from $n = 50k$ onward, with the F2 lead growing to +0.038 AUPR at $n = \text{full}$.

## 5. Conclusion and Limitations

On an out-of-time credit-abuse evaluation, we compare a learned sequence representation against engineered aggregates of the same credit events. Each feature set is evaluated under two classifiers — XGBoost and an in-context tabular foundation model — and the four resulting configurations are compared with paired-bootstrap CIs. **H1**: the learned sequence representation significantly improves AUPR over the engineered-aggregate alternative under both backbones (p<0.05). **H2**: starting from a GBDT-on-engineered baseline, the representation-only upgrade beats the classifier-only upgrade on both AUPR and recall (p<0.05). The H1 gap attributes to *structure* — order, timing, and co-occurrence of individual events — that scalar aggregates discard. Internal experiments with additional classifiers (TabICL, TabM, RealMLP, Mitra) are consistent with the reported findings: H1 and H2 both hold directionally on each. **Limitations.** Results cover a single platform and a single target. Multi-platform validation, longer event windows, and a pretraining-objective ablation are future work. The practical takeaway for early-life rare-label credit-abuse scoring: **upgrade the feature representation before upgrading the classifier**.

# A. Architecture, pretraining, finetuning, stacking details, and feature sets

**Table A.1.** *Transformer backbone hyperparameters.*

| Item | Value |
| --- | --- |
| Layers | 6 |
| Heads | 8 |
| $d_{\text{model}}$ | 128 |
| FFN width | 512 |
| Dropout (pretrain) | 0.3 |
| Event-type vocab | 13 |
| Max seq length | 192 |
| Time2Vec dim | 33 |
| Positional encoding | Sinusoidal |

**Table A.2.** *Self-supervised pretraining hyperparameters.*

| Item | Value |
| --- | --- |
| Objectives | Next-event CE (13) + temporal ranking |
| $\lambda$ (joint loss) | 0.3 |
| TR head | Bilinear, dim 128 |
| TR margin $\delta$ | 0.5 |
| TR negatives $K$ | 3 |
| TR sampling $p$ | 0.3 (truncated geometric, $\Delta \geq 2$) |
| Epochs | 6 |
| Batch size | 1024 |
| Optimizer | AdamW, lr $10^{-4}$, wd 0.05 |
| LR schedule | Exponential, $\gamma = 0.95$/epoch |
| Top-K seqs/user | 5 |

**Table A.3.** *Supervised finetuning hyperparameters.*

| Item | Value |
| --- | --- |
| Objective | Focal BCE, last-position head |
| Focal $\gamma, \alpha$ | 3.0, 0.25 |
| Initialization | Pretrained backbone |
| Neg downsampling | 10:1 |
| Epochs | 3 |
| Batch size | 1024 |
| Backbone lr | $10^{-4}$ |
| Head lr | $5 \times 10^{-4}$ |
| Weight decay | 0.01 |
| LR schedule | Exponential, $\gamma = 0.95$/epoch |
| Dropout | 0.2 |

**Downstream classifier configurations.**

**XGBoost (Table 1).** `binary:logistic`, depth 6, $\eta = 0.1$, 300 rounds, subsample 0.8, colsample 0.8, `min_child_weight=10`, `scale_pos_weight=30`, `tree_method=hist`.

**TabPFN in-context (Table 1).** TabPFN v3.

**TabPFN fine-tuned (App. C.3).** TabPFN v3 with end-to-end FT. Two regimes: full window at natural prevalence; *negative-downsampled* (all 268 positives + 9,732 sampled negatives, 1:36 prevalence).

**Mitra fine-tuned (App. C.3).** Mitra mixed-prior FT, *negative-downsampled* regime.

**TabPFN-Enhanced (App. C.3).** TabPFN v3, `use_enhanced=True`.

# B. Data, protocol, and evaluation

**Study population.** Mid-tenure applicants only — credit-file age between 36 and 60 months (months-on-file, external-bureau signal). All splits below are restricted to this segment.

**Time windows and cohort split.** Training window `2024-01-01` → `2025-06-01`; OOT test window `2025-10-01` → `2026-03-01` with a 120-day observation gap to realize the credit-abuse label. User cohorts are formed by disjoint hashing of applicant identifier (independent of the time axis). Split statistics are in Table B.1.

**Subsampling protocol.** When the in-context classifier's row cap is binding, a uniform-random training subsample is drawn (no class stratification). The same subsample is used for a cap-matched XGBoost reference so the classifier-vs-classifier contrast is read apples-to-apples.

**Bootstrap protocol.** Resamples preserve original $n_+/n_-$. Marginal CIs (Table 1 / Appendix C.1) are 95 % two-sided percentile intervals (2.5th–97.5th percentiles, 1000 iterations). Paired-bootstrap differences (Table 2 / Appendix C.2) are computed on the same resampled rows once per iteration; H1 / H2 are directional, so we report **one-sided 95 %** CIs (5th–95th percentiles) and call a comparison p < 0.05 when the lower bound exceeds zero.

# C. Extended results

This appendix decomposes the body's `OOT-merged` results by cohort (Tables C.1, C.2) and reports alternative classifiers tested as ablations (Table C.3).

We evaluate three alternatives to the in-context TabPFN baseline reported in Table 1: fine-tuned TabPFN, fine-tuned Mitra, and TabPFN-Enhanced. TabPFN's `use_preprocessing=True` flag is verified to be a no-op on our purely numeric features and reproduces the in-context baseline bitwise; we omit it from the table.

For the FT variants we report two training regimes. **full** trains on the full mid-tenure window at natural class prevalence (1 : 542) — the conventional FT regime. **negative-**

*Table B.1. Split statistics.*

| Split | Cohort × window | $n$ rows | $n$ positives | Positive rate |
|---|---|---|---|---|
| `train` | seen × training | 145,593 | 268 | 0.18 % |
| `OOT-out-of-cohort` | unseen × OOT | 27,411 | 47 | 0.17 % |
| `OOT-in-cohort` | seen × OOT | 109,215 | 170 | 0.16 % |
| **`OOT-merged`** (primary) | both × OOT | **136,626** | **217** | **0.16 %** |

*Table C.1. Main panel decomposed by cohort. AUPR and recall@0.5% on the three OOT splits with 95% class-stratified bootstrap CIs.*

| Metric | Classifier | Feature set | OOT-out-of-cohort (47+) | OOT-in-cohort (170+) | OOT-merged (217+) |
|---|---|---|---|---|---|
| AUPR | XGBoost | Engineered | 0.121 [0.051, 0.231] | 0.059 [0.034, 0.104] | 0.068 [0.044, 0.106] |
| | TabPFN (in-context) | Engineered | 0.129 [0.064, 0.244] | 0.094 [0.060, 0.145] | 0.097 [0.066, 0.142] |
| | XGBoost | Sequence | 0.187 [0.092, 0.311] | 0.139 [0.089, 0.201] | 0.147 [0.105, 0.201] |
| | TabPFN (in-context) | Sequence | 0.155 [0.078, 0.277] | 0.137 [0.088, 0.201] | 0.141 [0.095, 0.197] |
| recall@0.5% | XGBoost | Engineered | 0.149 [0.043, 0.277] | 0.118 [0.071, 0.176] | 0.124 [0.083, 0.171] |
| | TabPFN (in-context) | Engineered | 0.191 [0.085, 0.319] | 0.182 [0.118, 0.235] | 0.184 [0.134, 0.235] |
| | XGBoost | Sequence | 0.234 [0.106, 0.362] | 0.229 [0.171, 0.294] | 0.230 [0.180, 0.286] |
| | TabPFN (in-context) | Sequence | 0.298 [0.170, 0.426] | 0.206 [0.153, 0.271] | 0.230 [0.175, 0.281] |

*Table C.2. Paired-bootstrap differences by cohort. $\Delta = A - B$ AUPR. One-sided 95% paired-bootstrap [LB, UB] (5th–95th percentiles); **bold** indicates LB > 0 ($p < 0.05$).*

| Cell A | Cell B | OOT-out-of-cohort | OOT-in-cohort | OOT-merged |
|---|---|---|---|---|
| XGBoost + Sequence | XGBoost + Engineered | **+0.067 [+0.010, +0.127]** | **+0.080 [+0.043, +0.114]** | **+0.079 [+0.046, +0.107]** |
| TabPFN + Sequence | TabPFN + Engineered | +0.026 [−0.030, +0.069] | **+0.044 [+0.011, +0.078]** | **+0.044 [+0.015, +0.070]** |
| XGBoost + Sequence | TabPFN + Engineered | +0.059 [−0.014, +0.112] | **+0.045 [+0.008, +0.079]** | **+0.051 [+0.018, +0.078]** |
| TabPFN + Sequence | XGBoost + Sequence | −0.033 [−0.075, +0.013] | −0.002 [−0.027, +0.029] | −0.006 [−0.028, +0.019] |
| TabPFN + Engineered | XGBoost + Engineered | +0.008 [−0.044, +0.081] | **+0.035 [+0.008, +0.063]** | **+0.028 [+0.003, +0.053]** |

**downsampled** keeps all 268 abuse-positive training rows and uniform-samples 9,732 negatives, yielding a 10,000-row training set at 1 : 36 prevalence — the rare-event FT regime, with ratios chosen so each FT mini-context contains ~40 positives in expectation. FT-TabPFN learning rate is selected separately for each (feature set, regime) configuration from $\{1 \times 10^{-5}, 5 \times 10^{-6}, 2 \times 10^{-6}\}$ by validation ROC-AUC on a 10 % class-stratified holdout of training (OOT untouched during selection); we report results averaged over 3 finetuning seeds.

Under the held-out-validation LR-selection protocol, fine-tuning does not improve over in-context TabPFN on three of the four FT-TabPFN configurations (Engineered × full, Engineered × negative-downsampled, Sequence × full); pooled 95 % CIs all contain the in-context point estimate. The exception is Sequence × negative-downsampled, with AUPR 0.170 vs in-context 0.141 — a borderline lift consistent with TabPFN's documented strength in small-data regimes, since the 10,000-row class-balanced subsample provides ~40 positives per FT mini-context versus ~3 at natural prevalence.

H2 remains intact when fine-tuned TabPFN is admit-

*Table C.3. In-context TabPFN alternatives: AUPR and recall@0.5% on `OOT-merged` (n_pos = 217). FT-TabPFN rows: 3-seed mean point estimates with hierarchical-bootstrap pooled 95 % CIs (3 seeds × 1000 class-stratified iters). Other rows: single-seed 95 % class-stratified bootstrap CIs.*

| Classifier | Feature set | Regime | AUPR | recall@0.5% |
|---|---|---|---|---|
| Fine-tuned TabPFN | Engineered | full | 0.092 [0.046, 0.181] | 0.160 [0.101, 0.235] |
| Fine-tuned TabPFN | Engineered | negative-downsampled | 0.098 [0.062, 0.151] | 0.131 [0.083, 0.184] |
| Fine-tuned Mitra | Engineered | negative-downsampled | 0.075 [0.050, 0.109] | 0.134 [0.092, 0.180] |
| TabPFN-Enhanced | Engineered | full | 0.091 [0.058, 0.134] | 0.189 [0.143, 0.244] |
| Fine-tuned TabPFN | Sequence | full | 0.084 [0.038, 0.151] | 0.161 [0.101, 0.226] |
| Fine-tuned TabPFN | Sequence | negative-downsampled | 0.170 [0.119, 0.228] | 0.221 [0.157, 0.286] |
| Fine-tuned Mitra | Sequence | negative-downsampled | 0.069 [0.047, 0.104] | 0.171 [0.124, 0.226] |
| TabPFN-Enhanced | Sequence | full | 0.127 [0.085, 0.176] | 0.212 [0.161, 0.267] |

ted as a classifier-upgrade candidate. Its comparison axis is XGBoost+Sequence (AUPR 0.147) versus in-context TabPFN+Engineered (AUPR 0.097): the XGBoost+Sequence value is independent of the FT-TabPFN protocol, and as established above fine-tuning does not improve TabPFN's performance on engineered features (the FT-TabPFN+Engineered configurations land at 0.092 and 0.098, both within the in-context CI), so the TabPFN+Engineered value is also unaltered. The Sequence × negative-downsampled configuration that exceeds 0.147 sits off the H2 axis.

Per-configuration seed-of-finetune variance (across-seed std 0.007–0.034) is comparable to between-LR mean spread, indicating LR is not a clean recipe dial on this dataset. TabPFN-Enhanced and fine-tuned Mitra both trail in-context TabPFN on each feature set: TabPFN-Enhanced × Sequence sits at 0.127 (vs 0.141), and fine-tuned Mitra well below at 0.069 / 0.075.

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
