# OpenReview forum: "Learned Sequence Representations over Raw Credit Events for Credit-Abuse Scoring"
_ICML.cc/2026/Workshop/FMSD — FMSD @ ICML 2026 Poster_

### Official Review · Reviewer_TS2F · 2026-05-20
**Learned Sequence Representations over Raw Credit Events for Credit-Abuse Scoring**

**Rating:** 8
**Confidence:** 4

**Review:**

The paper demonstrates on a credit-abuse scoring dataset that switching from hand-engineered features to learned representations provides a larger performance boost than switching classifiers from XGBoost to TabPFN.

The paper is relevant to this workshop, as it addresses heterogeneous event streams and employs event embeddings pretrained using next-event-type prediction and temporal ranking tasks.

---

### Official Review · Reviewer_ewuc · 2026-05-21
**Learned Sequence Representations over Raw Credit Events for Credit-Abuse Scoring**

**Rating:** 3
**Confidence:** 5

**Review:**

## Summary
The paper studies whether a learned representation over raw credit-event sequences beats engineered scalar aggregates of the same events for credit-abuse scoring, a rare early-life default label. The authors pretrain a 6-layer causal Transformer over a 13-event-type vocabulary using two self-supervised objectives (next-event-type CE and a temporal-ranking margin loss), then finetune with focal BCE on the abuse label. The 128-d last-position embedding is concatenated with a 71-d engineered "complement" (external bureau / applicant context), giving a 199-d feature vector. A controlled 2×2 design crosses {sequence, engineered} × {XGBoost, in-context TabPFN} on an out-of-time evaluation (~136k applicants, 217 positives). Two hypotheses are tested with paired one-sided bootstrap: **(H1)** sequence > engineered under either classifier, and **(H2)** the representation upgrade dominates the classifier upgrade as a single-factor pick from the (XGBoost, engineered) baseline. Both are confirmed at p<0.05 on AUPR and recall@0.5%, with H2 holding at production training scale (≥50k rows).

## Strengths
The 2×2 grid in which both feature vectors share the same 71-d engineered complement and differ only in how the within-platform event stream enters isolates the variable of interest, and matching the 199-d cardinality of both sets controls for raw input dimensionality. The statistical protocol (class-stratified bootstrap CIs for marginals and paired one-sided bootstrap for directional comparisons in Tables 1, 2, C.2) matches the rare-label, directional structure of the H1/H2 claims; reporting [LB, UB] brackets with p<0.05 declared as LB>0 is appropriate for the claim being made. The OOT split decomposes into in-cohort and out-of-cohort with a 120-day observation gap, separating temporal generalization from cohort generalization. The data-efficiency sweep (Fig. 2) shows H1 holds at every n and locates the threshold at which H2 emerges (n ≥ 50k), which qualifies the headline rather than masking the regime dependence. Appendix C.3 evaluates fine-tuned TabPFN, fine-tuned Mitra, and TabPFN-Enhanced under both prevalence regimes with held-out-validation LR selection; FT-TabPFN does not surpass the in-context baseline on three of four configs, supporting the headline configuration choice.

## Weaknesses
The representation-choice motivation is missing throughout. Time2Vec is selected without comparison against alternatives (sinusoidal time-only, absolute-timestamp embedding, learned delta-bucket embeddings, no time encoding); no ablation tells the reader whether Time2Vec is load-bearing or arbitrary, and the same gap applies to the per-event-type MLP tokenization, the choice of two pretraining objectives (next-event-type CE plus temporal ranking) over either alone, and the truncated geometric negative-sampling distribution. A more fundamental experimental gap is the absence of a *combined* feature-set experiment: the 2×2 grid pits learned sequence and engineered aggregates as substitutes, but never tests them together. A (sequence-embedding ⊕ within-platform engineered aggregates ⊕ external complement) condition would directly answer the deployment-relevant question of whether the learned representation *adds* signal over engineered features or merely *replaces* them; as written, the paper conflates the two cases. There is no public-dataset evaluation, and the proprietary dataset is also under-described: the platform, label-prevalence breakdown, bureau-covariate provenance, and basic schema are not given, so the result is both unreproducible and uninspectable, and a public-data anchor (Amex default, Kaggle Home Credit, IBM TabFormer transactions) would have addressed both concerns at once. The engineered baseline is plausibly a strawman: the 128 within-platform aggregates are described only as drawn "from the full internal pool" with no statement of how they were selected (gain ranking, domain curation, random subset), so it is impossible to tell whether they represent the platform's best-effort hand-engineering or a deliberately weak comparison; if the production-pipeline aggregates differ from these 128, that fact alone would substantially reframe the H1 lift. The H2 framing also has a structural issue: XGBoost and in-context TabPFN deliver near-equal performance on the same feature set, so the "classifier upgrade" axis of H2 is essentially flat, and "representation > classifier" is partly an artifact of the classifier arm barely moving rather than the representation arm being uniquely powerful; H2 should be discussed in those terms rather than as a clean single-factor preference. Beyond these, the result rests on 217 OOT positives on a single platform with a single backbone; PRAGMA, FinLangNet, and nuFormer are cited but not run as baselines; a no-pretraining ablation (random-init backbone, finetune only) and a non-Transformer sequence baseline (LSTM over the same tokenization) are absent, so the contribution of the proposed pretraining recipe cannot be isolated from "any sequence model." Taken together, the missing representation ablations, the missing combined-feature experiment, the missing public-data anchor, the under-described engineered baseline, and the flat classifier axis make the contribution thin even at workshop scope, not only at the main track.

---

### Official Review · Reviewer_sSKb · 2026-05-22
**Learned Sequence Representations over Raw Credit Events for Credit-Abuse Scoring**

**Rating:** 3
**Confidence:** 5

**Review:**

## Summary
This work investigates whether directly encoding the sequential structure of credit events via a pretrained Transformer improves credit-abuse prediction compared to conventional hand-crafted scalar aggregates of those same events. Credit abuse is framed as a rare early-life default concentrated in applicants whose credit-file age falls between 36 and 60 months. The authors design a compact tokenization layer (per-event-type MLPs over a 13-type vocabulary, Time2Vec temporal encoding, and learned event-type embeddings summed at each position) fed into a 6-layer causal Transformer (8 heads, d=128, FFN=512, max length 192). Self-supervised pretraining combines masked next-event-type cross-entropy with a temporal-ranking margin objective (truncated geometric negative sampling, K=3, margin 0.5). After finetuning with focal BCE (gamma=3, alpha=0.25, 10:1 negative downsampling), the last-position hidden state yields a 128-d embedding concatenated with a shared 71-d engineered complement (external bureau + applicant context), forming a 199-d feature vector. The parallel engineered baseline replaces this 128-d embedding with 128 within-platform scalar aggregates to match dimensionality. A controlled 2x2 factorial — {sequence, engineered} x {XGBoost, in-context TabPFN v3} — evaluated on an out-of-time split (training 2024-01 to 2025-06; OOT 2025-10 to 2026-03, 120-day observation gap, ~136k applicants, 217 positives) yields: **(H1)** sequence-based features significantly beat engineered aggregates under both classifiers on AUPR and recall@0.5% (one-sided paired bootstrap, p<0.05), and **(H2)** the representation upgrade dominates the classifier upgrade as a single-factor investment from the (XGBoost, engineered) baseline, with this effect emerging at n>=50k training rows.

## Strengths
- **Clean factorial isolation.** Both feature vectors share the same 71-d external complement and differ only in how within-platform events enter (embedding vs. aggregates). Matching dimensionality at 199-d removes a confound that many representation-learning papers leave open.
- **Rigorous statistical protocol.** Class-stratified bootstrap CIs for marginals (Table 1) and one-sided paired-bootstrap brackets for directional H1/H2 claims (Table 2) are well-suited to the low-prevalence, directional-hypothesis setting. Reporting 5th–95th percentile intervals and declaring significance when LB > 0 is both transparent and appropriate.
- **Cohort-decomposed OOT evaluation.** Splitting OOT into `in-cohort` (seen applicants in future window) vs. `out-of-cohort` (unseen applicants in future window) disentangles temporal generalization from user-cohort generalization — a finer-grained test than a single temporal hold-out.
- **Data-efficiency analysis (Fig. 2).** Sweeping n in {5k, 10k, 25k, 50k, 100k, full} at five seeds shows H1 holds at every regime, while H2 only activates at n>=50k, giving a nuanced practical message: invest in representation first, but the marginal dominance over a classifier switch requires sufficient downstream data.
- **Thorough alternative-classifier ablation (Appendix C.3).** Fine-tuned TabPFN, Fine-tuned Mitra, and TabPFN-Enhanced are tested under both natural-prevalence and negative-downsampled regimes with validation-LR-selected configurations. That fine-tuned TabPFN fails to beat in-context TabPFN on 3/4 configurations justifies the main-body choice and addresses a natural reviewer concern.
- **The wide-and-deep stacking paradigm** (citing Cheng et al. 2016) makes the architecture straightforward to integrate into an existing production pipeline: freeze the pretrained backbone, extract embedding, concatenate, and hand off to an existing tree scorer.

## Weaknesses
- **No representation-choice ablations.** Time2Vec is adopted without comparison to simpler temporal encodings (sinusoidal-only, learned delta-bucket, absolute timestamp, or no time encoding at all). Likewise, the per-event-type MLP tokenization, the choice of two pretraining losses versus either alone, the truncated-geometric sampling distribution versus uniform, and the margin value delta=0.5 are all unablated. The reader cannot tell which design decisions are load-bearing.
- **Missing combined-feature condition.** The 2x2 grid frames sequence and engineered features as substitutes, but in deployment the obvious next question is whether they are complementary. A (sequence-embedding + within-platform aggregates + external complement) arm would clarify whether the learned representation adds signal beyond what hand-engineering already captures or merely recapitulates it. Without this, the H1 "replaces" interpretation cannot be separated from "adds on top of."
- **Opaque engineered baseline.** The 128 within-platform aggregates are described only as drawn "from the full internal pool." Whether they are the production team's best-performing set, a gain-ranked top-128 subset, or a random sample is never stated. If the actual production pipeline uses a richer or differently-selected feature set, the headline H1 lift may overstate real-world gains.
- **H2 is partly an artifact of a flat classifier axis.** XGBoost and in-context TabPFN deliver near-identical performance on the same feature set (Table 1: 0.068 vs 0.097 on engineered, 0.147 vs 0.141 on sequence). Because the classifier upgrade barely moves the needle either way, "representation > classifier" is mechanically easy to satisfy; the paper should discuss this rather than framing H2 as evidence of general representation dominance.
- **No public-dataset anchor or reproducibility pathway.** The platform, label schema, bureau covariate provenance, and even the country are anonymized. A parallel evaluation on a public credit dataset (Amex Default, Kaggle Home Credit, or IBM TabFormer transactions) would let the community validate the approach independently without revealing proprietary data.
- **No non-Transformer or no-pretraining baselines.** An LSTM over the same tokenization and a randomly-initialized Transformer (finetune only, no pretraining) are absent. Without these, the contribution of the specific pretraining recipe cannot be distinguished from "any temporal architecture on raw events."
- **Narrow evaluation on a single binary target.** Only one platform and one abuse label are tested. The 217 OOT positives (0.16% prevalence) produce wide bootstrap intervals; the approach's generality to other rare-event credit labels (e.g., first-party fraud, bust-out, synthetic identity) or other platforms is unknown.
- **PRAGMA, FinLangNet, and nuFormer are cited in Related Work but never benchmarked.** These are the most directly comparable Transformer-on-credit-events systems. Running at least one would ground the claim against a meaningful state-of-the-art reference rather than only against engineered aggregates.
- **No calibration analysis.** Credit scoring models are deployed as risk scores; AUPR and recall@threshold report discrimination but say nothing about whether the predicted probabilities are well-calibrated. A reliability diagram or Brier-score comparison would strengthen the practical relevance.